# The Route of Motor Recovery in Stroke Patients Driven by Exoskeleton-Robot-Assisted Therapy: A Path-Analysis

**DOI:** 10.3390/medsci9040064

**Published:** 2021-10-26

**Authors:** Loris Pignolo, Rocco Servidio, Giuseppina Basta, Simone Carozzo, Paolo Tonin, Rocco Salvatore Calabrò, Antonio Cerasa

**Affiliations:** 1S’Anna Institute, 88900 Crotone, Italy; pina.basta@gmail.com (G.B.); s.carozzo@isakr.it (S.C.); patonin18@gmail.com (P.T.); antonio.cerasa@irib.cnr.it (A.C.); 2Department of Cultures, Education and Society, University of Calabria, 87036 Rende, Italy; rocco.servidio@unical.it; 3IRCCS Centro Neurolesi Bonino-Pulejo, 98124 Messina, Italy; roccos.calabro@irccsme.it; 4Institute for Biomedical Research and Innovation (IRIB), National Research Council of Italy (CNR), 98164 Messina, Italy; 5Pharmacotechnology Documentation and Transfer Unit, Preclinical and Translational Pharmacology, Department of Pharmacy, Health Science and Nutrition, University of Calabria, 87036 Rende, Italy

**Keywords:** stroke, exoskeleton-robot-assisted therapy, Fugl–Meyer, path analysis

## Abstract

*Background*: Exoskeleton-robot-assisted therapy is known to positively affect the recovery of arm functions in stroke patients. However, there is a lack of evidence regarding which variables might favor a better outcome and how this can be modulated by other factors. *Methods*: In this within-subject study, we evaluated the efficacy of a robot-assisted rehabilitation system in the recovery of upper limb functions. We performed a path analysis using a structural equation modeling approach in a large sample of 102 stroke patients (age 63.6 ± 13.1 years; 61% men) in the post-acute phase. They underwent 7 weeks of bilateral arm training assisted by an exoskeleton robot combined with a conventional treatment (consisting of simple physical activity together with occupational therapy). The upper extremity section of the Fugl–Meyer (FM-UE) scale at admission was used as a predictor of outcome, whereas age, gender, side of the lesion, days from the event, pain scale, duration of treatment, and number of sessions as mediators. *Results:* FM-UE at admission was a direct predictor of outcome, as measured by the motricity index of the contralateral upper limb and trunk control test, without any other mediating factors. Age, gender, days from the event, side of lesion, and pain scales were independently associated with outcomes. *Conclusions*: To the best of our knowledge, this is the first study assessing the relationship between clinical variables and outcomes induced by robot-assisted rehabilitation with a path-analysis model. We define a new route for motor recovery of stroke patients driven by exoskeleton-robot-assisted therapy, highlighting the role of FM-UE at admission as a useful predictor of outcome, although other variables need to be considered in the time-course of disease.

## 1. Introduction

Several systematic and meta-analytic reviews have confirmed that robotic-assisted devices boost motor recovery in patients with stroke, mainly considering the upper limb intervention [1,2,3]. Early research on robotic therapy for the upper limb was based on end-effector robots, which hold the patient’s hand or forearm at one point and generate forces at the interface. Recently, this field of study has shifted towards exoskeleton devices, which overcome many of the inherent limitations of end-effector robots [4]. Compared to conventional therapy, exoskeletons have the potential to provide intensive and repetitive rehabilitation sessions with a longer duration, regardless of the skills and effort of the therapist [3,5].

The Automatic Recovery Arm Motility Integrated System (ARAMIS) is a concept robot and prototype for the neurorehabilitation of the paretic upper limb developed by the S. Anna Institute, Crotone, Italy [6,7]. ARAMIS was designed with two computer-controlled, symmetric, and interacting exoskeletons, which compensate for the inadequate strength and accuracy of the paretic arm movements and the effect of gravity during rehabilitation. The basic idea is to exploit proprioceptive inputs using passive, repetitive, interactive, high-intensive bilateral movement training, which has been demonstrated to enhance motor recovery in stroke patients [8,9,10,11]. This device has been widely validated [6,7,12] for conventional neurorehabilitation approaches, demonstrating a high degree of upper limb recovery, as assessed by the Fugl-Meyer (FM) scale [13]. Moreover, we recently demonstrated that lesion load in the superior region of the corona radiata, internal capsule, and putamen were significantly associated with recovery of the upper limb, as defined by the FM scores in stroke patients who underwent motor rehabilitation with ARAMIS [14].

The FM is a well-known performance-based impairment index designed to assess motor functioning, sensation, balance, and joint functioning in patients with post-stroke hemiplegia. Overall, FM, together with the functional independence measure (FIM) scale [15], is considered the most reliable clinical scale employed to unravel motor recovery after robotic-related treatments in stroke patients [1].

Despite this large amount of evidence confirming the effectiveness and robustness of robotic-assisted rehabilitation in promoting motor recovery, there is a lack of evidence regarding which variables might predict a better outcome and how this can be modulated by other factors. Recently, Coupar et al. [16], after reviewing 58 prognostic studies of upper limb recovery, showed that the initial severity of motor impairment or function is the most important predictor for upper limb recovery after stroke, although this conclusion did not consider exoskeleton-robot-assisted therapy at all. For this reason, there is a need to transform rehabilitation treatments into “intelligent rehabilitation” processes to provide valid support for clinicians, given the difficulty to distinguish among several variants in the clinical practice.

In the big data era, the employment of artificial intelligence (AI) algorithms has provided a unique opportunity to improve clinical practice [17,18]. Their implementation has been tested in every level of stroke care to improve diagnosis, techniques of treatment, as well as rehabilitation [17]. The vast majority of these algorithms employed knowledge-based methods that utilize knowledge to construct diagnostic/prognostic models and perform inference processes. In the neurorehabilitation realm, the application of AI is in its relative infancy, since there are several issues to be solved before translation to clinical practice. Indeed, the neurorehabilitation cycle includes four main steps (assessment, assignment, intervention, and evaluation), which are influenced by a plethora of demographical, psychological, neurological, and environmental factors that are not yet well-defined [19]. Thus, before applying advanced automated approaches to the complex neurorehabilitation cycle, it is first necessary to identify how the process of motor recovery is developed and which factors might mediate this effect.

For this reason, we applied, for the first time, a path analysis model to motor recovery data of stroke patients undergoing robotic-assisted rehabilitation to determine which specific demographic and clinical factors could mediate the relationship between the initial severity of motor impairment with upper limb recovery.

## 2. Materials and Methods

### 2.1. Enrollment

We enrolled patients who met the criteria for the first attack of sub-cortical ischemic stroke recruited at the Sant’Anna Rehabilitation Center. From an initial cohort of 224 subacute hemiplegic patients, we enrolled only those who fulfilled the following criteria: (i) unilateral first-ever ischemic or hemorrhagic supratentorial stroke, (ii) ability to follow verbal instructions, and (iii) right-handed patients. Exclusion criteria were (1) bilateral impairment; (2) severe sensory deficits in the paretic upper limb; (3) pregnancy, epilepsy, cognitive impairment (Mini-Mental State Evaluation, MMSE < 24) or behavioral dysfunction that would influence the patient’s ability to participate in the treatment; (4) botulinum toxin injections or other medication influencing upper-limb function; (5) inability to provide informed consent.

All participants gave written informed consent. The study was approved by the Ethical Committee of the Regione Calabria, according to the Helsinki Declaration.

All patients completed an extensive series of clinical tests that were administered by an experienced physician. In detail, the degree of disability during daily living activities was assessed with the Barthel Index [20], and the motor strength of the contralateral/ipsilateral upper limbs was assessed with the motricity index (MI) [21]. Patients’ synergistic motor control of the paretic arm was assessed with the upper extremity (UE) section of the FM (FM-UE) [22]. Further measures included the functional independence measure (FIM) and the trunk control test (TCT) [23].

### 2.2. ARAMIS Hard/Software Structure

The robotic-assisted rehabilitation performed through the ARAMIS device has been described in previous papers [6,7,12,14]. The ARAMIS framework is a fully integrated set of software that enables the therapist to program and manage the rehabilitation procedures. Briefly, the robotic platform includes two fully motorized 6 DOF symmetric exoskeletons (Figure 1). Kinematic and dynamic data are continuously acquired and stored by the control system, which evaluates the weight torque and compensates for it by controlling each upper limb posture and the strength delivered by the patient to the exoskeleton. Movements are, therefore, supported by a drive motor, which adjusts its strength on a step-by-step basis.

Each exoskeleton can record (motion capture) the movements of the unaffected arm and the patient is requested to replicate every single movement by the paretic arm in synchronous or asynchronous modalities, depending on the exercise typology or training program, with continuous compensation for the paretic arm’s inadequate strength and accuracy. ARAMIS-assisted rehabilitation is possible in three different modalities: (1) asynchronous—the patient wears both exoskeletons and uses the unaffected arm to perform pre-programmed exercises that are replicated by the paretic arm supported by its exoskeleton; (2) synchronous—the unaffected arm paces the movements to be replicated synchronously and with the same physical characteristics (such as strength, acceleration, range, and speed) by the exoskeleton hosting the paretic arm; (3) active-assisted—when the patient is not able to carry out a movement, the robot supports the arm strength against gravity, thus replicating movements executed by the unaffected arm.

### 2.3. Design and Procedure

We used a within-subject design divided into four main stages. The first stage was based on the recruitment of the patients (see inclusion criteria reported above). Physiotherapists, as well as data entry assistants, were blinded to all phases of the study. In the second stage, the eligible stroke patients underwent an MRI examination at baseline (T0). In the third stage, participants underwent a rehabilitation program consisting of a validated protocol of exoskeleton-robot-assisted activities [6,7,12] combined with an additional 4–5 h per week of conventional therapy [24,25,26], in agreement with Italian norms on the treatment of stroke patients. The conventional activities were carried out by a (blinded) expert therapist and consisted of occupational therapy exercises together with passive/active mobilization of upper and lower limbs, trunk control, standing, and ambulation. The ARAMIS protocol for rehabilitation included 60 min sessions daily over periods not exceeding 7 weeks. Both single and multiple movements were scheduled. In the first 2–3 weeks of treatment, all subjects performed a series of asynchronous exercises, where the paretic arms repeated each of the exercises 20 times for a total of 200 repetitions per session (Table 1). In the following 2–3 weeks, the asynchronous exercises were gradually reduced to 100 per session and replaced by synchronous exercises (100/session), with the total number remaining unchanged. The rehabilitation sessions in the active-assisted modality began following adequate motor recruitment of the upper limb where necessary as stipulated in the FM-UE scale modified by Lindmark and Hamrin (total score > 70) [22], which continued to the end of scheduled treatment. All patients started rehabilitation periods at the same time. Finally, at the end of the 7 week training period, participants were given a blinded motor assessment by an external clinician, using the same protocol as at baseline (T1).

### 2.4. Data Analysis

SPSS 25 package was used to perform the preliminary statistical analyses. Therefore, a path analysis using a structural equation modelling (SEM) was implemented with the software Mplus 7.04 [26]. Path analysis is the simplest case of SEM, since the model contained only observed variables. As recommended by Hu and Bentler [27], multiple indices were used to assess the fit of the model (adopted cut-offs in brackets): the chi-square (χ^2^) test value with the associated *p*-value (*p* > 0.05), comparative fit index (CFI ≥ 0.90), Tucker–Lewis index (TLI ≥ 0.90), root mean square error of approximation (RMSEA ≤ 0.06) and its 90% confidence interval, and standardized root mean square residual (SRMR < 0.08). Parameter estimation for the proposed path-SEM model was computed with the maximum-likelihood (MLM) parameter with standard errors and a mean-adjusted chi-square test statistic that were robust to non-normality, as Maydeu-Olivares [28] suggested. The MLM chi-square test statistic is also referred to as the Satorra–Bentler (S-B) chi-square.

## 3. Results

### 3.1. Clinical Data

One hundred and two-stroke patients (age: 63.6 ± 13.1; 61% male; with left side lesions) completed all phases of the rehabilitation protocol (Figure 2). Clinical information is summarized in Table 1. Briefly, the duration of therapy lasted 44.4 ± 20.8 days, and patients underwent 22.2 ± 10.6 sessions. The hemiplegic side was mainly localized in the left hemisphere (68%). After the treatment, most patients showed evident motor recovery in all clinical outcomes: (A) FM-UE scale average global score: from 36.7 ± 20.7 at baseline to 63.26 ± 25.5 after treatment); (B) MI average global scores: from 49.9 ± 40.5 at baseline to 70.5 ± 20.5; from 61.74 ± 42.6 at baseline to 74.9 ± 33.4 after treatment, respectively, for contralateral and ipsilateral upper limbs; (C) FIM average global score: from 51.9 ± 24.3 at baseline to 83.5 ± 22.4 after treatment; (D) Barthel index average global score: from 23.8 ± 67.4 at baseline to 67.4 ± 26.3 after treatment; and (E) TCT average global score: from 36.1 ± 20.8 at baseline to 80.8 ± 18.7 after treatment.

### 3.2. Path Analysis

The results of the tested model provided a good fit to the data, χ^2^S-B (20, *N* = 34) = 24.74, *p* = 0.212, CFI = 0.974, TLI = 0.901, RMSEA = 0.080, 90% CI [0.000, 0.178], and SRMR = 0.067 (Figure 3). Specifically, the current results indicated a positive association between FM-UE at admission and time from stroke (Post), β = 0.180, SE = 10, *p* < 0.05, which, in turn, leads to MI-I, β = 0.130, and SE = 0.05, *p* < 0.01. FM-UE at admission was directly (negatively) related to TCT, β = 0.435, and SE = 16, *p* < 0.01 and MI-C, β = 0.412, SE = 15, and *p* < 0.01. Conversely, FM-UE at admission was positively associated with Vas, β = 0.260, SE = 14, and *p* < 0.05, which, in turn, leads to MI-I, β = 0.297, SE = 12, and *p* < 0.05.

However, there were no statistically significant mediation effects. Considering the mediating factors, we found that they are independently related to outcome. (a) Gender is related to Barthel index scores (females are associated with greater gains); (b) age influences FM-UE at discharge, Barthel index (younger stroke patients have better performance), and TCT (elderly patients have better scores). Finally, the side of lesion impacts on MI scores of the controlateral/ipsilateral upper limbs (patients with right lesions showed better recovery).

## 4. Discussion

The establishment of an adequate prognosis is one of the most important clinical targets aimed at increasing the efficiency of stroke services and at reducing costs. Robotics therapies have been shown to be beneficial in improving arm function [16], and a plethora of outcome measures for clinical motor neurorehabilitation have been proposed [16]. In this study, for the first time, we sought to evaluate the strict relationship between outcome measures using a path analysis approach. We provide a new relationship model to shed new light on the complex interaction between motor severity and recovery in stroke patients who underwent robotic-assisted therapy, considering several demographic and clinical variables. We found that FM-UE at admission is a useful prognostic factor to predict the recovery of the trunk and the motricity of both upper limbs in stroke patients receiving robotic neurorehabilitation. In particular, those patients with great impairment in the upper paretic limb at admission had greater gains later, as measured by TCT and MI scores. Otherwise, the other traditional outcome measures (such as Barthel index, FM-UE at discharge) are independently influenced by a series of clinical and demographical factors (Figure 3).

The initial assessment of the FM-UE scores of the paretic arm has been demonstrated to be one of the most optimal predictors of outcome in patients with stroke [29]. In particular, in a recent international consensus-based core set for clinical motor rehabilitation, it has been established that FM-UE scores are one of the most reliable measures to be considered in the management of stroke patients and for predicting the time course of disease [30]. Path analysis confirmed the high prognostic value of this factor, but specifically for the motor gain measured by TCT and MI scales. The lack of mediational effects could be dependent upon the other intermediate variables, which are not tested in the present study. Indeed, age at injury is recognized as an important prognostic factor of the disease progression. It has also been demonstrated that this demographical factor is strongly associated with activities of daily living after stroke [31]. König et al. [32] claimed that age, together with the National Institutes of Health Stroke Scale score, is sufficient to correctly predict survival and functional recovery after 3 months. However, the predicting value of age on stroke rehabilitation continues to be controversial [33]. Our path analysis demonstrated that younger patients are, independently of an initial impairment, associated with better recovery, as measured by FM-UE and Barthel index measurements post-intervention. Gender has also been considered a marker of functional recovery. Recently, Poggesi et al. [34] demonstrated that, during the neurorehabilitation period, there is a substantial overlap of functional recovery in men and women after stroke. However, at older ages, the potential for recovery appeared better in women compared to men. Our data agree with this evidence, although it is important to highlight that a direct comparison among studies should consider the different statistical approaches employed to define the prediction value of a clinical variable to others.

However, the relationship between motor severity assessed in the post-stroke early phase and the outcome may also depend on the concomitant spontaneous recovery, presence of pain, and side of the lesion. VAS is generally used for the subjective measurement of mood, pain, and health status after stroke. Indeed, stroke survivors experience significant pain that may affect the effectiveness of motor rehabilitation, mainly when using exoskeleton-robot-assisted therapy. It has been demonstrated that this kind of rehabilitation approach may improve the perception of pain. Kim et al. [35] demonstrated that a prototype robot for shoulder rehabilitation improves hemiplegic shoulder pain and self-reported shoulder-related disability. Our path analysis agrees with this interesting finding, showing that patients with high chronic pain had greater gains, as measured by MI of the ipsilateral upper limb. On the contrary, those patients with great impairment in the upper paretic limb at admission had greater pain scores. Again, time from the event is generally considered as a common underlying intrinsic mechanism known as “spontaneous neurological recovery” [36]. The time course after stroke is characterized by larger improvements during the first weeks post-stroke, which non-linearly increased until the sixth month. This natural mechanism accounted for 40% of motor recovery alone [36]. Our path analysis showed that patients with a longer time from the event had a greater gain, as measured by MI of the ipsilateral upper limb. On the other hand, patients with great impairment in the upper paretic limb at admission had a long time from the event. Finally, it is widely recognized that stroke patients with lesions in the left hemisphere are characterized by more cognitive deficits (aphasia, neglect, apraxia) that could have affect neurorehabilitation goals [37]. Our data confirm that patients with right lesions are associated with more motor gains in both upper limbs as measured by MI.

We are not able to explain the reason why the number of training sessions does not predict the recovery, given that it is well known that neuroplasticity is influenced by repetition and intensity of the treatment. However, the sample was not so homogeneous, and this could have biased some results. Studies with larger samples comparing low-intensity vs. high-intensity robotic UL rehab, especially in patients in the chronic phase, are needed to pave the way for a better definition of common protocols in robotic rehabilitation following stroke.

The definition of the best predictors of outcome in stroke patients who underwent robotic-assisted therapy has a deep impact on long-term assistance. Indeed, after the neurorehabilitation period, the acquired recovery should be maintained overtime when patients return home. As stated by Wan et al. [38], the development of mobile healthcare robots is expected to dramatically improve healthcare services in the future. In this new field of study, it is mandatory to develop an integrated information architecture that will facilitate a remotely teleoperated health robot service at home, which may link with health professionals [39]. To define the core of this information architecture, it is important to eliminate uncertainty data or clinical outcomes. We retain that our work has the merit to provide a new way to define the relationships between clinical variables with outcomes, using path analysis that allows differentiating correlations into direct effect and indirect effects.

### Limitations

There are some limitations to this study. First is the lack of a useful control group for evaluating the impact of robotic-assisted rehabilitation for conventional treatment. Although we are aware of this limitation, it is important to bear in mind that the route of a motor recovery driven by conventional therapy has widely been demonstrated [6,7,12,14]. The main target of this study was to investigate which clinical variables may mediate the direct relationship motor severity at admission through motor recovery driven by the employment of an exoskeleton robotic-assisted system. Therefore, a path-SEM analysis was carried out to test the associations among the main variables of the study as well as to design a more complex model with direct and indirect effects. Indeed, studies in this field mainly apply regression approaches, which do not allow us to model causal mechanisms. The second limitation is related to the exclusion of other critical clinical variables that could have a mediator effect on the outcome, such as brain imaging information or EEG/EMG data [30,36].

## 5. Conclusions

To the best of our knowledge, this is the first study assessing the relationship between clinical variables and outcomes induced by robot-assisted rehabilitation with a path-analysis model. We define a new route for motor recovery of stroke patients driven by robot-assisted therapy, highlighting the role of FM-UE at admission as a useful predictor of the outcome, which should be considered before addressing stroke patients to exoskeleton-robot-assisted therapy of upper limbs.

## Figures and Tables

**Figure 1 medsci-09-00064-f001:**
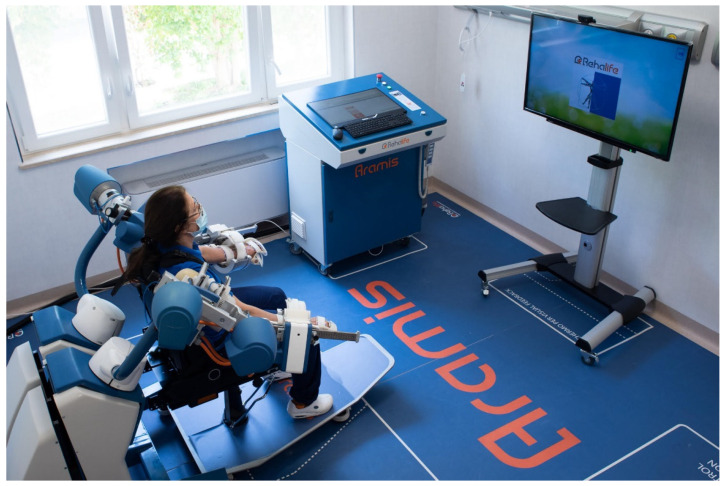
The robotic-assisted device called Automatic Recovery Arm Motility Integrated System (ARAMIS).

**Figure 2 medsci-09-00064-f002:**
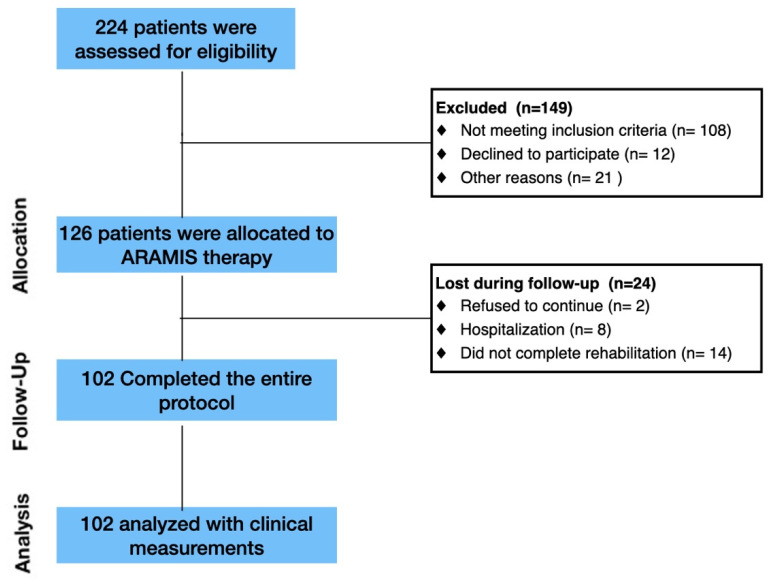
Flow diagram of participant recruitment and participation in the study: stroke patients participated in an individualized robotic-assisted neurorehabilitation program (ARAMIS system).

**Figure 3 medsci-09-00064-f003:**
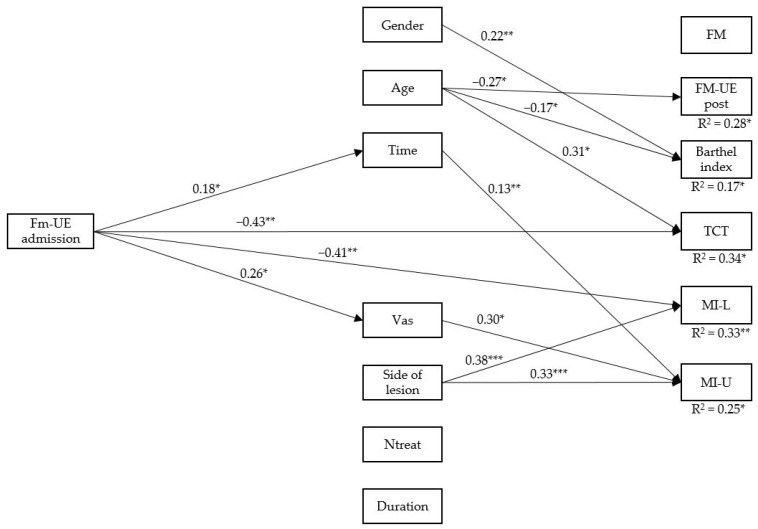
Final model with significant paths. For clarity, only the standardized significant relationships are depicted in the figure. * *p* < 0.05. ***p* < 0.01. *** *p* < 0.001. *FM-UE* = Fugl–Meyer; Time = time elapsed from stroke (days); Vas = visual analogue scale; *Ntreat* = number of treatments; Fim = functional independence measure; Trunk = trunk control test; MI-C = motricity index contralateral upper limb; MI-I = motricity index ipsilateral upper limb.

**Table 1 medsci-09-00064-t001:** Clinical and demographical characteristics of stroke patients.

**Variables**			
Number	102		
Gender (% male)	62%		
Age at injury (years)	63.6 ± 13		
Hemispheric lesion (% left)	68%		
Days from event	21.9 ± 15.1		
Length of stay IRU (days)	44.4 ± 20.8		
Number of sessions	22.7 ± 10.6		
VAS at admission	49.9 ± 25.1		
FM-UE at admission	36.7 ± 20.7		
	**Before Rehabilitation**	**After Rehabilitation**	**Delta Improvement**
FM-UE	36.7 ± 20.7	63.2 ± 25.5	+26.6 ± 18.7
FIM	51.9 ± 24.3	83.5 ± 22.5	+31.6 ± 20.2
Barthel-Index	23.8 ± 15.6	67.4 ± 26.2	+43.6 ± 23.1
TCT	36.1 ± 20.8	80.8 ± 18.7	+44.1 ± 17.7
MI-C	49.9 ± 40.4	70.5 ± 30.9	+20.5 ± 19.3
MI-I	61.74 ± 42.6	74.9 ± 33.4	+13.1 ± 20.3

## Data Availability

The datasets generated during and/or analysed during the current study are available from the corresponding author on reasonable request.

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
