# Peer review of "The Route of Motor Recovery in Stroke Patients Driven by Exoskeleton-Robot-Assisted Therapy: A Path-Analysis"

_medsci, 2021, doi:10.3390/medsci9040064_

Round 1
Reviewer 1 Report
The paper is clearly written and well structured. Authors have investigated the impact of demographic and clinical factors to upper limb recovery after exoskeleton-robot assisted therapy in a large sample of 102 post-acute stroke patients. The paper could be accepted in the current form.
Author Response
REPLY: We would like to thank this reviewer for these favorable comments.
Reviewer 2 Report
- Change in introduction [1-3[, in [1-3]
- It seems a god survey with sound openings for improvements, as also the authors mentioned. Looking forward for next results
In this case it looks like a fair research paper, presenting even unnecessary drawbacks that increase the limitation of the research, i.e. : not direct justification of the 7 weeks robotic training (knowing the literature and the project financing is clear); or therapy duration (44.4 ± 20.8 days) that is from single to double “unjustified” duration, having just a swift mention of the abandon rate. That may lead to the need of different classification and integration of the patients, but that may be another matter. Therefore I write in my review that the actual findings lead to new openings in the field.
Author Response
- Change in introduction [1-3[, in [1-3]
REPLY: Done
- It seems a god survey with sound openings for improvements, as also the authors mentioned. Looking forward for next results
In this case it looks like a fair research paper, presenting even unnecessary drawbacks that increase the limitation of the research, i.e. : not direct justification of the 7 weeks robotic training (knowing the literature and the project financing is clear); or therapy duration (44.4 ± 20.8 days) that is from single to double “unjustified” duration, having just a swift mention of the abandon rate. That may lead to the need of different classification and integration of the patients, but that may be another matter. Therefore I write in my review that the actual findings lead to new openings in the field.
REPLY: We thank this reviewer for this observation and general positive feeling about our study. Please consider that the aim of this study was not to change or improve a robotic-assisted therapy approach that has been demonstrated to provide a well-defined motor recovery in stroke patients admitted to IRU. But we need to understand which is the intimately link between clinical variables and outcomes providing a new relationship model to shed new light on the complex interaction between motor severity and recovery. Our scientific contribution relies on statistical improvement with future influence on clinical practice.
Reviewer 3 Report
This study tries to find the variables that might favour a better outcome on exoskeleton-robot asised therapy. The research topic is well motivated. But after reading the paper, this study only concludes that age, gender, days from the event, side of lesion and pain scales were independently associated with outcomes. The reviewer believes that this reusult is not impressive. Besides, the experimental results are not well orgonised in the format of, i.g. tables or bar charts. Last but not the least , typos (i.g. Line 37) and grammar errors should be avoided.
In general, this paper should be reorganised to include better result demonstration.
Author Response
- This study tries to find the variables that might favour a better outcome on exoskeleton-robot asised therapy. The research topic is well motivated. But after reading the paper, this study only concludes that age, gender, days from the event, side of lesion and pain scales were independently associated with outcomes. The reviewer believes that this reusult is not impressive. Besides, the experimental results are not well orgonised in the format of, i.g. tables or bar charts. Last but not the least , typos (i.g. Line 37) and grammar errors should be avoided.
Reply: Following the reviewer’s suggestion, we now provide a new version checking for typos and grammar errors.
- In general, this paper should be reorganised to include better result demonstration.
Reply: We regret that this reviewer considers our study not impressive. Please consider that this is the first time that a path-analysis approach has been applied to this field of study. This statistical approach allows differentiating correlations into direct effects and indirect effects. We provide a new relationship model to shed new light on the complex interaction between motor severity and recovery, considering several demographic and clinical variables, in stroke patients who underwent robotic-assisted therapy. Our main finding is that FM-UE at admission was a direct predictor of the outcome as measured by the Motricity Index of the contralateral upper limb and Trunk Control Test. The fact that motor recovery (as measured by FIM or Barthel Index) is, instead, not specifically related to FM-UE at admission but to other single clinical or demographical variables is another piece of knowledge useful for clinicians. To the best of our knowledge, this is the first time that such a direct relationship has been demonstrated. Moreover, the provided enrollment approach, power statistical analysis, clinical and demographic description of participants (Table 1), as well as path analysis (Figure 3) have been conducted using standard internationally accepted criteria. We would appreciate if this reviewer could better specify which part of the results is not well described.
Reviewer 4 Report
In this within-subject study, we evaluated the efficacy of a Robot-assisted rehabilitation system in the recovery of upper limb functions. We performed a path analysis using a structural equation modelling approach in a large sample of 102 stroke patients (age 63.6 ± 13.1 years; 61% men) in the post-acute phase. The topic of this paper is relevant and interesting; The overall feeling of the reviewer, however, is that the technical contribution of this paper is limited. Some issues the authors need to address are as follows.
(1) What's new in this work compared with previous work? The authors need to elaborate not only why this topic is important, but also the relevant recent researches and existing problems, as well as what they have done to advance this area, so as to make the paper more valuable.
(2) In section 2.2, there already exists a wealth of research about applications of stroke patients driven by Exo-
skeleton-Robot Assisted Therapy in the related areas. Have you proposed any algorithms or improvements over the exisiting methods?
(3) The references are not enough and some are outdated. Some more recent references are needed to reflect the timeliness of this paper, such as : Cognitive computing and wireless communications on the edge for healthcare service robots, Computer Communications, 2020. Faster R-CNN for multi-class fruit detection using a robotic vision system, Computer Networks, 2020.
Author Response
- (1) What's new in this work compared with previous work? The authors need to elaborate not only why this topic is important, but also the relevant recent researches and existing problems, as well as what they have done to advance this area, so as to make the paper more valuable.
REPLY: Following the reviewer’s suggestion the final part of the introduction has been modified and improved to better present the novelty of this study.
- (2) In section 2.2, there already exists a wealth of research about applications of stroke patients driven by Exo-skeleton-Robot Assisted Therapy in the related areas. Have you proposed any algorithms or improvements over the exisiting methods?
REPLY: As stated in the introduction, the aim of this work is to define a new relationship model to shed new light on the complex interaction between motor severity and recovery, considering several demographic and clinical variables, in stroke patients who underwent robotic-assisted therapy. For this reason, we employed a well-validated robotic device with a well-defined protocol. We didn’t need to change or improve a robotic-assisted therapy approach which has been demonstrated to provide a well-defined motor recovery in stroke patients. But we need to understand which is the intimate link between clinical variables and outcomes. Our scientific contribution relies on statistical improvement with a deep influence on clinical practice and on the application of “intelligent rehabilitation processes” in the future.
3) The references are not enough and some are outdated. Some more recent references are needed to reflect the timeliness of this paper, such as : Cognitive computing and wireless communications on the edge for healthcare service robots, Computer Communications, 2020. Faster R-CNN for multi-class fruit detection using a robotic vision system, Computer Networks, 2020.
REPLY: we would like to thank this reviewer for suggesting these impressive works. The Introduction and discussion have been reformulated accordingly.